# Review on Inflammation Markers in Chronic Kidney Disease

**DOI:** 10.3390/biomedicines9020182

**Published:** 2021-02-11

**Authors:** Tadej Petreski, Nejc Piko, Robert Ekart, Radovan Hojs, Sebastjan Bevc

**Affiliations:** 1Department of Nephrology, Clinic for Internal Medicine, University Medical Centre Maribor, Ljubljanska Ulica 5, 2000 Maribor, Slovenia; tadej.petreski@ukc-mb.si (T.P.); nejc.piko@ukc-mb.si (N.P.); radovan.hojs@ukc-mb.si (R.H.); 2Department of Internal Medicine and Department of Pharmacology, Faculty of Medicine, University of Maribor, Taborska Ulica 8, 2000 Maribor, Slovenia; 3Department of Dialysis, Clinic for Internal Medicine, University Medical Centre Maribor, Ljubljanska Ulica 5, 2000 Maribor, Slovenia; robert.ekart@ukc-mb.si

**Keywords:** inflammation markers, chronic kidney disease, cytokines, chemokines, cell adhesion molecules

## Abstract

Chronic kidney disease (CKD) is one of the major health problems of the modern age. It represents an important public health challenge with an ever-lasting rising prevalence, which reached almost 700 million by the year 2017. Therefore, it is very important to identify patients at risk for CKD development and discover risk factors that cause the progression of the disease. Several studies have tackled this conundrum in recent years, novel markers have been identified, and new insights into the pathogenesis of CKD have been gained. This review summarizes the evidence on markers of inflammation and their role in the development and progression of CKD. It will focus primarily on cytokines, chemokines, and cell adhesion molecules. Nevertheless, further large, multicenter studies are needed to establish the role of these markers and confirm possible treatment options in everyday clinical practice.

## 1. Introduction

Chronic kidney disease (CKD) represents a condition with reduced renal function, which is defined by glomerular filtration rate (GFR) < 60 mL/min per 1.73 m^2^ or with present markers of kidney damage or both for more than 3 months and with implications for health. Markers of kidney damage are defined by Kidney Disease Improving Global Outcomes (KDIGO) guidelines from 2012 as albuminuria, urine sediment abnormalities, electrolyte, and other abnormalities due to tubular disorders, abnormalities detected by histology, structural abnormalities detected by imaging, and history of kidney transplantation. CKD should be categorized into categories according to GFR ranging from G1–G5 (GFR in G1 ≥ 90, G2 60–89, G3a 45–59, G3b 30–44, G4 15–29, and G5 ≤ 15 mL/min/1.73 m^2^), and albuminuria should be accounted for in categories from A1–A3 (albumin excretion rate in A1 < 30, A2 30–300, and A3 > 300 mg/24 h) [1]. The pathophysiology of CKD is extremely complex with its clinical course dependent on a broad spectrum of different etiologies all leading towards kidney failure. The main causes of CKD in all high-income and middle-income countries are diabetes mellitus (DM) and arterial hypertension (AH) [2]. The global prevalence of all-stage CKD recorded in 2017 was 697.5 million or 9.1%, and 1.2 million people died from CKD in 2017 [3]. Patients are often asymptomatic or have non-specific symptoms such as loss of appetite, pruritus, and fatigue, so diagnosis is usually achieved by estimation of GFR using several available equations, or less frequently by measurement of GFR via exogenous markers. Additionally, a diagnosis can be made by performing a kidney biopsy, which can commonly show glomerular sclerosis, tubular atrophy, and interstitial fibrosis [2]. One of the hallmarks of CKD resulting in its development, progression, and complications is persistent, low-grade inflammation [4,5].

Inflammation is a natural and necessary body response to different stimuli. It is responsible for the migration of immune system cells to the stimuli target site following a series of steps facilitated and coordinated by cytokines, chemokines, and acute-phase proteins. In the acute setting, this provides a resolution of the problem and enables the return to the status quo. Chronic inflammation on the other hand can lead to tissue damage and fibrosis. As such it has been associated with numerous diseases, CKD included [6]. We are only beginning to understand this very fine-tuned mechanism and research is still ongoing. Until now, several different pathways of inflammation response have been discovered, such as p38 mitogen-activated protein kinase (p38 MAPK), interleukin-6 (IL-6)/Janus kinase (JAK)/signal transducer and activator of transcription 3 (STAT3), and phosphoinositide 3-kinase (PI3K) [7].

This review will summarize recent evidence on markers of inflammation in CKD. It will focus primarily on cytokines, such as interleukins (ILs), tumor necrosis factor (TNF), interferon (IFN), and transforming growth factor (TGF), chemokines, and cell adhesion molecules (CAMs) (Figure 1). 

## 2. Cytokines

Cytokines are small proteins (15–20 kDa) involved in the development and activity of the immune system. They have an important effector and messenger role with involvement in autocrine, paracrine, and endocrine signaling [8,9]. Their short half-life suggests they are usually rapidly eliminated to ensure limited bioactivity but can exhibit systemic effects if necessary. The acute phase is usually short term and sufficient to resolve the injury. Persistent inflammation, either due to prolonged stimulation by biological, chemical, or physical stimuli, or inappropriate reaction against self-antigens, however, can prove to be problematic, as it can lead to development of chronic inflammation [6]. Possible association of cytokines with diabetic nephropathy was already observed almost 30 years ago [10].

### 2.1. Interleukins

Interleukins are a group of cytokines first seen to be expressed by leukocytes. Due to receptor complexes that interact with various interleukins, they have been divided into families, the most important being: IL-1 superfamily, IL-6 family, IL-10 family, IL-12 family, and IL-17 family [8,9,11,12,13].

#### 2.1.1. Interleukin 6

The IL-6 family cytokines consist of IL-6, IL-11, ciliary neurotrophic factor (CNTF), leukemia inhibitory factor (LIF), oncostatin M (OSM), cardiotrophin 1 (CT-1), cardiotrophin-like cytokine (CLC), and IL-27, which are four-helical proteins and interact with the receptor subunit glycoprotein 130 kDa (gp130). Their signal transduction is dominated by signal transducer and activator of transcription 3 (STAT3) activation, except IL-27, which signals via STAT1 [8].

IL-6 is one of the most studied cytokines in kidney disease known for its proinflammatory actions, including B-cell stimulation and induction of hepatic acute phase proteins, but also included in metabolic, regenerative, and neural processes as well [8,14]. It is mostly known for its role in autoimmune disorders such as rheumatoid arthritis, where serum levels of IL-6 can raise thousand-fold [8]. The Chronic Renal Insufficiency Cohort (CRIC) study has demonstrated that elevated levels of IL-6 in CKD patients resulted in a greater decline of estimated glomerular filtration rate (eGFR) but lose their significance after adjustment for albuminuria [15]. One of the main mechanisms responsible for IL-6′s role in CKD is atherosclerosis. In dialysis patients, IL-6 is associated with increased central aortic stiffness measured with carotid-femoral pulse wave velocity (cfPWV) [16]. When looking at IL-6 in South African CKD patients, higher levels of IL-6 were associated with advanced carotid plaques and were present with -174G/C polymorphism in the IL-6 gene [17]. Although, a recent meta-analysis has shown no significant correlation of -174G/C polymorphism with the risk of developing end-stage kidney disease (ESKD) [18].

Other possible roles of IL-6 are the increase in fibroblast growth factor 23 (FGF23) transcription in acute kidney injury (AKI) and CKD, which is associated with increased morbidity and mortality [19]. Additionally, it contributes to the development of anemia in juvenile CKD in an iron-independent path, causing renal fibrosis and altering the erythropoietin axis [20]. Its role in erythropoiesis has also been observed in adults. Higher levels of IL-6 are independently associated with reduced responsiveness to erythropoiesis-stimulating agents (ESA) therapy in hemodialysis patients [21]. Furthermore, studies on mice suggest its role in the development of hypertension and end-organ damage, with possible treatment options [22].

However, some studies show conflicting results regarding IL-6′s role on renal function. The authors Salimi et al. tested several markers of inflammation in the inChianti study, where IL-6 levels at baseline or their change during observation were not consistently significantly associated with renal function in older adults [23].

Nevertheless, studies on drugs that target IL-6 or IL-6 receptors are being conducted. A prior proof of concept study on lupus nephritis patients has failed to elucidate any efficacy of IL-6 inhibition with sirukumab in addition to regular treatment [24]. However, recently Pergola et al. reported a phase 1/2 randomized control trial (RCT) assessing the effects of ziltivekimab, a novel anti-IL-6 ligand antibody, in hemodialysis patients with a polymorphism for heightened susceptibility of IL-6′s inflammatory effects. Treated patients had improved markers of inflammation, increased serum albumin, and reduced use of ESAs [25]. Moreover, case reports of tocilizumab use in rheumatoid arthritis with AA amyloidosis and CKD showed renal function preservation and lowering of proteinuria [26]. Studies on specific targeting of IL-6 trans-signaling by using the *sgp130Fc* protein show a potentially novel strategy of renal fibrosis treatment by suppressing STAT3 activation without systemic IL-6 inhibition [8,27].

#### 2.1.2. Interleukin 1 and 18

The IL-1 family consists of 11 members, 7 pro-inflammatory agonists (IL-1α, IL-1β, IL-18, IL-36α, IL-36β, IL-36γ, and IL-33) and 4 antagonists (IL-1RA, IL-36Ra, IL-37, and IL-38) with anti-inflammatory properties [9,28]. They have a fundamental role in the induction and modulation of the innate immune response, such as nondirected phagocytosis, complement activation, or activation of innate immune receptors, where an imbalance can lead to exaggerated inflammatory responses. Recent evidence suggests they have an additional role in trained innate immunity and autoimmune disorders [9,28,29]. IL-1β is one of the best characterized members, which binds to IL-1 receptor and results in immune activation and fever. It is also involved in the generation of T helper 17 cells, which are important in the pathogenesis of autoimmune diseases such as psoriasis and rheumatoid arthritis. Its transcription is activated by toll-like receptor (TLR) or IL-1 receptor activation or TNF, additionally caspase-1, an inflammasome, is required to cleave it into the mature form [9].

Several studies investigating the benefit of IL-1 inhibition on kidney function have been reported. In a post-hoc analysis of the REDHART study, Buckley et al. concluded that IL-1 could be a potential key mediator of inflammation in heart failure and CKD as they have observed a fall in inflammation markers after the use of *anakinra*, a recombinant human IL-1 receptor antagonist. However, no change in renal function was observed after 12 weeks of treatment [30]. Similarly, Nowak et al. did not witness an improvement in CKD mineral and bone disorder or physical function after 12 weeks of treatment with *rilonacept*, an IL-1 inhibitor [31]. However, they carried out another study targeting vascular function, where *rilonacept* use in CKD was associated with improved brachial artery flow-mediated dilation and reduced systemic inflammation [32]. Additionally, in the CANTOS trial, they used *canakinumab*, a human monoclonal antibody targeting IL-1β, in CKD patients to observe a reduction in major cardiovascular events, with no effects on kidney function [33]. Hung et al. recently observed that an IL-1 inhibition in CKD patients improves the anti-inflammatory and antioxidative properties of the high-density lipoprotein (HDL)-containing fraction of plasma [34].

IL-18 is a similar cytokine to IL-1 as it must be cleaved by caspase-1 for activation and uses the same pathway to activate the nuclear factor kappa-light-chain-enhancer of activated B cells (NF-kB) and to induce inflammatory mediators. It is mostly secreted from myeloid cells and its binding to the IL-18 receptor causes an increase in IFN gamma production, which is why it is increased in autoimmune diseases such as lupus erythematosus, psoriasis, and inflammatory bowel disease [9,35]. However, it seems that it is more specific to renal pathophysiology, especially in diabetic nephropathy, where its expression is positively correlated to albuminuria and progression of the disease [36]. Due to its role in diabetic nephropathy, its inhibition could serve as a potential therapeutic target [37]. A systematic review and meta-analysis have shown that urine IL-18 could potentially be used as a biomarker for AKI [38].

#### 2.1.3. Inflammasome

Inflammasomes are large molecular complexes with several functions but mostly known for activating the proteolytic caspase 1 pathway [39]. In CKD it not only mediates the inflammatory response, but is associated with pyroptosis, mitochondrial regulation, and myofibroblast differentiation as well [40]. Both IL-1 and IL-18 are activated by the NOD- (nucleotide oligomerization domain), LRR- (leucin-rich repeat), and pyrin domain-containing protein 3 (NLRP3) inflammasome. Its activation in renal phagocytes and podocytes results in a release of IL-1 and IL-18, which causes inflammation during CKD [41]. Researchers have observed NLRP3′s role in renal fibrosis, diabetic nephropathy, obesity-related kidney disease, chronic glomerulonephritis, immunoglobulin A nephropathy (IgAN), crystal-related nephropathy, and hyperhomocysteinemia-induced renal injury [42]. Raised levels of plasma oxalate can lead to inflammasome activation and recent data shows that lowering plasma oxalate could prevent or mitigate CKD [43]. Similar findings have been attributed to use of allopurinol, which has shown to decrease renal inflammation by limiting activation of NLRP3 inflammasome [44]. Interestingly, studies have shown that the NLRP3 has several inflammasome-independent functions on CKD pathogenesis through regulation of cell apoptosis and tissue remodeling, as well as mitochondrial regulation [40,41,45,46]. Nephrocalcinosis and hyperoxaluria are two known conditions that involve NLRP3, where TGF receptor signaling and macrophage polarization play a role [47]. Furthermore, several studies have suggested NLRP3 inhibition to be a novel treatment target with a possible effect on renal function, renal fibrosis, hypertension, atherosclerosis, and DM [40,48,49,50,51]. Additionally, NLRP3 inhibition could attenuate obesity-related kidney disease [52]. A novel agent *pterostilbene*, an analog of resveratrol with autophagy-inducing effects, has shown its renal protective effects in an invitro study of urate nephropathy by down-regulating the NLRP3 inflammasome [53].

#### 2.1.4. Other Interleukins

IL-17A is a member of the IL-17 family, which consists of IL-17A through IL-17F, that participates in tissue inflammation, where it aids especially against cutaneous bacterial and fungal infections. Moreover, it is involved in autoimmune and inflammatory diseases, such as psoriasis, ankylosing spondylitis, and psoriatic arthritis, where IL-17 inhibition is a possible treatment option with agents *secukinumab* and *ixekizumab*, as well as in the pathogenesis of AKI and CKD [13,54]. Recent evidence suggests its inhibitory role in tumor growth factor beta (TGF-β) induced renal interstitial fibrosis [55]. Several studies show that its increased levels are present in lupus nephritis and IgAN, along with crescentic glomerulonephritis [56,57,58,59]. Dedong et al. reported that it could be used as a biomarker for estimating lupus nephritis disease activity and a prediction marker for treatment response [60].

IL-10 is a member of the IL-10 family, next to IL-19, IL-20, IL-22, IL-24, and IL-26, that has a known role as an anti-inflammatory mediator. It is produced mainly by immune cells, as well as tissue epithelial cells. Its part has been studied especially in tissue fibrogenesis, which also includes renal fibrosis [11,61,62]. IL-10′s interaction with other signaling molecules is being studied as a new potential antifibrotic therapeutic strategy [63]. Due to its anti-inflammatory actions, it improves vascular and renal function in preclinical hypertension studies [64]. A recent meta-analysis on IL-10 gene polymorphisms suggests protective effects of some single nucleotide polymorphisms (SNPs) on the risk of developing diabetic nephropathy in type 2 DM [65].

### 2.2. Tumor Necrosis Factor

TNF is another cytokine released by immunologic cells, tumor necrosis factor alpha (TNF-α) specifically by monocytes/macrophages. The TNF ligand family is coded by 18 known human genes; its relative the TNF/nerve growth factor (NGF) is coded by an additional 29 genes. However, not only ligands are important in tissue homeostasis, but their tumor necrosis factor receptors (TNFR) as well. Approximately 30 members of TNFR have been identified, which are type I transmembrane proteins that can be classified into three groups: death receptors, TNFR-associated factor, and decoy receptors [66,67]. These ligands and receptor families have several known functions: control of cell death, orchestration of inflammation, tissue modeling, control of adaptive immunity, and several less understood mechanisms tied to control of brain function, such as perception, fever, anorexia, sleep, etc. [67,68]. They also have an important role in autoimmunity, cancer, infectious disease, and graft-versus-host disease [67]. One of the possible ways it can act on CKD is by raising FGF23, which regulates phosphate homeostasis, causes mineral and bone disorder, and is associated with all-cause mortality of CKD patients [69].

Studies show that TNF-α levels are higher in diabetic patients compared to healthy people, with its levels rising with worsening diabetic nephropathy [70]. Higher soluble TNFR’s were also associated with diabetic kidney disease, where it independently predicted incident cardiovascular disease and mortality [71]. Additionally, their circulating TNFR levels indicated eGFR decline, as well as progression to ESRD [72]. A study on mice has revealed that TNF-α blockade in diabetic nephropathy confers kidney protection visible by reduction of albuminuria, serum creatinine, histopathologic changes, and macrophage recruitment to kidneys [73].

As TNF’s role in diabetic nephropathy has been established, other renal diseases were investigated as well. Oh et al. have reported that higher levels of circulating TNFR are associated with renal function decline as well as worse histopathologic findings in patients with IgAN. Additionally, baseline values predicted subsequent renal progression [72]. TNFR’s effect on declining kidney function was also reported in the Heart and Soul study [74]. Similar findings were reported for TNF-α, where higher serum levels proved to be a potential biomarker for evaluating disease severity as they were linked with proteinuria and renal function decline [75]. Interestingly, however, TNF-α inhibitors have been identified as a potential cause for IgAN, which was reported for use of *adalimumab*, but not for *infliximab* [76]. Nonetheless, their use in patients with rheumatoid arthritis from the KOBIO registry did not influence any change in renal function [77]. Furthermore, a study on rats has shown that a combination of *empagliflozin,* a sodium-glucose co-transporter 2 (SGLT2) inhibitor, and *infliximab* prevented renal fibrosis to a greater extent than monotherapy, which could present a promising therapeutic option [78].

Genetic studies have revealed that gene polymorphisms of TNF-α are risk factors for the development of nephrotic syndrome among Egyptian children and are linked with steroid resistance [79]. Additionally, some TNF gene polymorphisms in Mexican patients were associated with systemic lupus erythematosus (SLE) and lupus nephritis [80].

### 2.3. Interferons

IFNs are a class of cytokines first identified due to their property to interfere with viral replication in the host. They have a versatile function ranging from the regulation of immune response, which is associated with autoimmunity, to oncogenesis. They are divided into three types. Type I, such as interferon alpha (INF-α) and interferon beta (IFN-β), are expressed by innate immune cells; type II, also termed interferon gamma (IFN-γ) is expressed by natural killer cells and T lymphocytes; and type III, also termed interferon lambda (IFN-λ) or initially IL-28 and IL-29, are distributed in tissue and act on epithelial surfaces. Type I IFNs bind to the IFN alpha and beta receptor subunit 1 and 2 (IFNAR1 and IFNAR 2), which signal through recruitment of tyrosine kinase 2 (TYK2) and JAK1, respectively. The same signal cascade is activated in IFN type III signaling, however, IFN- λ binds to IL28-R and IL-10R2. IFN-γ binds to the IFN-γ receptor 1 and 2 (IFNGR1 and IFNGR2), which signals through JAK1 and JAK2, respectively [81].

IFN’s role in CKD is less well understood. Undoubtedly it plays an important part in several extrarenal viral infections such as hepatitis C infection, which can also trigger membranoproliferative glomerulonephritis, however, it was also observed in IgAN, lupus nephritis, and renal vasculitis [82]. Grzegorzewska et al. reported that plasma IFN-λ levels in hemodialysis patients are associated with a better response to hepatitis B virus (HBV) vaccine and are independently associated with survival [83].

Several case reports have been described, where interferon was used as a treatment option, mostly for non-renal disease. A patient with immunoglobulin M (IgM) nephropathy presented with nephrotic syndrome and achieved long-term remission with IFN-α [84]. However, plasma levels of IFN-α were reported higher in patients with IgAN by Zheng et al., where they were associated with proteinuria and histologic changes [85]. Similarly, genetic studies reported an association of serum levels of IFN-γ and IgAN and suggested that IFN-γ polymorphisms may be involved in the development and progression of IgAN in the Chinese population [86]. Furthermore, a case report of a patient with hypereosinophilia syndrome who was treated with *recombinant IFN-α-2b* described he developed progressive renal failure and nephrotic proteinuria 1 year after IFN treatment. Cytokine therapy discontinuation led to a reversal of kidney injury [87]. Similar findings were observed in a patient with multiple sclerosis (MS), where therapy with IFN-β led to the development of nephropathy with nephrotic syndrome [88]. These findings can somewhat be explained with a study by Migliorini et al., who reported that INF-α and IFN-β both have a synergistic, yet different, effect on podocytes and parietal epithelial cells that causes glomerulosclerosis [89]. Due to known systemic adverse effects of IFN administration, some researchers have tried a different approach, using specific drug targeting on mice models. They targeted platelet-derived growth factor receptor β (PDGFRβ)-expressing myofibroblasts using IFN-γ, which attenuated renal fibrosis in obstructive nephropathy [90].

### 2.4. Transforming Growth Factor Beta

TGF-β is a cytokine involved in the regulation of cell proliferation. Regardless of its name, it can either stimulate or inhibit cell growth and is critical for maintaining immune cell homeostasis. Additionally, it is implicated in the pathogenesis of several diseases such as connective tissue disorders, fibrosis, and cancer. During research of the genome sequence, 33 proteins have been recognized, which now comprise the TGF-β family. They have different receptors and intracellular effectors named Smad proteins, which mediate intracellular signaling and result in regulation of expression of target genes. Gene mutations associated with TGF-β regulation and release are implicated in connective tissue disorders and skeletal disease [91,92,93].

Some CKDs such as membranous and diabetic nephropathy frequently result in ESRD due to renal fibrosis. One of the major regulators appears to be TGF-β, which induces epithelial to mesenchymal transition, up-regulates matrix protein synthesis, inhibits matrix degradation, and alters cell-to-cell interaction [94,95]. One of the pathways that is regulated by TGF-β and is involved in tissue fibrosis by scar formation by myofibroblasts is TGF-β/Mothers against decapentaplegic homolog 3 (SMAD3), additionally, non-canonical pathways such as src kinase, epidermal growth factor receptor (EGFR), JAK/STAT3, and p53 collectively drive the fibrosis process [96,97].

A meta-analysis revealed that elevated serum levels of TGF-β are present in diabetic patients, which poses a high risk for nephropathy [98]. Therefore, a study on mice evaluated the use of pyrrole-imidazole (PI) polyamide, promoter inhibitors targeting TGF-β1, which resulted in improvement of podocyte injury [99]. However, a phase 2 study tried to address the effect of TGF-β1 monoclonal antibody (LY2382770) use in patients with diabetic nephropathy on the renin–angiotensin–aldosterone system (RAAS) inhibitor treatment. The treatment has not resulted in slowing renal function decline [100]. Nonetheless, potential therapeutic targets for progressive renal fibrosis involved in the TGF-β/SMAD3 pathway recently discovered are long non-coding ribonucleic acid (lnc-RNA) Erbb4-IR and lnc-TSI [101,102]. TGF- β/SMAD3-driven fibrosis also represents the pathophysiologic basis of obstructive nephropathy. Recently, a small molecule called petchiether A was identified as a potential inhibitor of SMAD3 phosphorylation, which protects against renal inflammation and fibrosis [103]. A further molecular mechanism is the nuclear factor erythroid 2-like 2 (Nrf2)/SMAD7 pathways, which was targeted recently by *bardoxolone*, an activator of NRF-2, and resulted in amelioration of tubular necrosis and interstitial fibrosis in mice [104].

## 3. Chemokines

Chemokines (the term is derived from chemotactic cytokines) are a family of small peptides (8 to 12 kDa) that mediate various processes such as chemotaxis, leucocyte degranulation, hematopoiesis, and angiogenesis. They function by interacting with the cell surface G-protein coupled receptors (GPCRs) and are involved in several pathologic disorders such as cancer, infectious disease, and atherosclerosis. One of recent strategies to tackle these diseases has therefore been the direct inhibition of GPCRs but development of effective chemokine antagonists is elusive. Depending on the spacing of disulfide bridges between peptides, they are classified into four subfamilies: the CXC subfamily (α-subfamily), CC subfamily (β-subfamily), C or XC subfamily (γ-subfamily), and CX3C subfamily (δ-subfamily) [105,106].

### 3.1. CXC Subfamily

As genome-wide association studies (GWASs) revealed an association of locus in the CXCL12 (CXC ligand 12), also known as stromal cell-derived factor 1 alpha and beta (SDF-1α and β), region with myocardial infarction (MI), a study was done to assess this clinically. In CKD patients from the CRIC study, higher plasma levels of CXCL12 were associated with cardiovascular disease risk factors and incident MI and death [107]. Another study by Klimczak–Tomaniak reported an association of higher CXCL12 levels with left ventricular mass and 24 h blood pressure in CKD patients [108]. However, Chen et al. investigated the role of CXCL12/CXCR4 (CXC receptor 4) pathways in renal vascular development and observed that local CXCL12/CXCR4 signaling preserves microvascular integrity and prevents renal fibrosis. Additionally, this effect was observed after treatment with an angiotensin converting enzyme (ACE) inhibitor [109]. Ribeiro et al. reported that CXCL12 and CXCL8, also known as IL-8, could be used as a marker of vascular damage and dysfunction in uremia [110]. Similar findings were reported by Bouabdallah et al., where they observed that CXCL8 could have a role in vascular calcification in uremia by a mechanism of endothelial cells and vascular smooth muscle cells interaction [111]. CXCL8, alongside with CXCL1, also known as growth-related oncogene-α (GRO-α), and CXCL6, also known as granulocyte chemotactic protein (GCP), were significantly higher in patients with nephrotic syndrome compared to controls and their levels decrease with remission, which could suggest that they play a role in the pathogenesis of nephrotic syndrome [112]. Another chemokine named CXCL16, a scavenger receptor for oxidized low-density lipoprotein (LDL), was reported by Lin et al. to be independently associated with a change in renal function in all-stage CKD [113]. Additionally, it appears to be a marker of renal injury in patients with type 2 DM, where it is independently predictive of microalbuminuria [114,115] and its urinary levels could reflect the degree of interstitial fibrosis and tubular atrophy in patients with advanced diabetic kidney disease [116]. It appears to also play a role in the etiopathogenesis of preeclampsia, where its blockade could pave the way to the discovery of new treatments of preeclampsia [117]. CXCL10, also known as interferon-gamma-inducible protein 10 (IP-10), has shown its value in lupus patients. Urinary levels of CXCL10 were correlated with renal activity score, 24-h proteinuria, and SLE disease activity index (SLEDAI) score. This was also proven for serum CXCL10 and CCL2 levels, however, only urinary levels of CXCL10 indicated renal activity [118,119].

### 3.2. CC Subfamily

Plasma CCL2, also known as monocyte chemoattractant protein 1 (MCP-1), increases with lowering renal function, however, it appears to be an independent risk factor for death caused by other factors than atherosclerosis in patients with CKD [120]. Nonetheless, CCL2 levels were associated with dyslipidemia in hemodialysis patients [121]. CCL2 baseline values have also shown to be significantly associated with incident CKD [122]. Additionally, higher values of CCL2 reflect tubular injury in patients with diabetic nephropathy [123]. Gene polymorphism studies demonstrated that CCL2 polymorphisms (MCP-1 2518 A > G) may be associated with the risk of developing CKD [124]. Serum and urinary CCL2 were evaluated in pediatric CKD patients compared to controls. Higher levels were observed in CKD patients, with a difference in urinary levels between different CKD etiologies; patients with glomerular disease had higher levels than those with anomalies of the kidney and urinary tract [125]. Additionally, fractional excretion of CCL2 as a marker of inflammation was shown to precede tubular dysfunction in pediatric CKD patients [126]. CCL18, also known as the pulmonary and activation-regulated chemokine, was shown to drive renal inflammation in ANCA-associated crescentic glomerulonephritis and could serve as a biomarker for disease activity and relapse [127].

In biopsy specimens of patients with IgAN, CCL2, and CCL5, also known as regulated upon activation normal T-cell expressed and secreted (RANTES), as well as CCR5 were up-regulated and could play a role in renal fibrosis [128]. The possible role of CCL2 in renal fibrosis and worsening renal function was also highlighted following a systematic review [129]. Interestingly, CCL2 has shown kidney protective properties during acute inflammatory response after renal ischemic/reperfusion injury in mice [130]. Additionally, CCL5, known for its role in the recruitment of macrophages and T lymphocytes into injured tissues, has shown a protective role in a mice model of kidney injury by constraining the proinflammatory actions of CCL2 [131]. Nonetheless, a review of CCL2 regulation has shown that CCL2 blockade in several models of renal disease has ameliorated the disease [132].

### 3.3. CX3C Subfamily

CX3CL1, also known as fractalkine, has been involved in several kidney diseases, however, studies show conflicting results regarding its disease-promoting or protecting effect [133]. Nonetheless, recent mice studies show that it could play an essential role in lupus nephritis with the development of tubulointerstitial lesions through activation of the Wnt/β-catenin signaling pathway [134]. Additionally, in vitro studies further evaluated the CX3CL1 and Wnt/β-catenin signaling pathway to find that it is associated with podocyte damage and could account for progressive AKI [135]. In Chinese patients with IgAN, plasma CX3CL1 levels were associated with renal function, proteinuria, and histologic changes [136]. Furthermore, they suggest a possible role in the pathogenesis and activity of lupus nephritis [137]. However, a study on mice showed that CX3CR1 inhibition in hypertension could promote hypertensive renal injury [138]. A study investigating CX3CR1 V249I A/G gene polymorphism in CKD patients revealed its possible role in AH, DM, and atherosclerosis [139].

## 4. Cell Adhesion Molecules

CAMs have a central role in cell communication and connection. They are divided into five groups: integrins, selectins, cadherins, members of the immunoglobulin superfamily (IgSF), and others such as mucins. They differ between one another by structure and ligand binding, as well as the type of binding. Selectins are further divided into P- (platelet), E- (endothelial cells), and L- (leukocyte) selectins. Cadherins, derived from calcium-dependent adherent proteins, can be subdivided into classical cadherins (type I and II), protocadherins, and atypical cadherins. The largest family is the IgSF, which also includes the major histocompatibility complex (MHC) class I and II, and the proteins of the T cell receptor (TCR) complex, such as intracellular adhesion molecules (ICAMs) vascular cell adhesion molecules (VCAMs), mucosal addressin cell adhesion molecule (MAdCAM-1), and activated leukocyte cell adhesion molecule (ALCAM) [140]. The most important CAMs associated with atherogenesis identified until now are VCAM-1, ICAM-1, platelet endothelial cell adhesion molecule 1 (PECAM-1), E-selectin, and P-selectin. As they were identified, research was ongoing about designing treatment modalities by targeting specific CAMs to be able to deliver drugs directly to plaque site [141].

In CKD, several endothelial dysfunction biomarkers, such as ICAM, VCAM, and E-selectin are increased [142]. Additionally, the uremic milieu can increase the levels of endothelial microparticles, which are lined with several adhesion molecules, such as cadherins, ICAM, E-selectin, and integrins, which cause a proinflammatory state, and are related to the development of atherosclerosis [143]. Bevc et al. showed that intima-media thickness (IMT) correlates with serum VCAM-1 in hemodialysis patients [144]. A study by Feng et al. showed that decreasing eGFR is associated with higher VCAM as inflammation in CKD stimulates glomerular expression of VCAMs [145]. In adult-onset minimal change disease, levels of VCAM and E-selectin were increased and associated with severity of nephrotic syndrome [146]. A multicenter study revealed that IgAN and focal segmental glomerulosclerosis (FSGS) had higher levels of P-selectin than healthy controls [147]. However, in CKD patients with atrial fibrillation, levels of P-selectin and E-selectin did not correlate with eGFR [148]. In Asians with type 2 DM, VCAM, but not ICAM, was associated with prevalent diabetic kidney disease and renal function decline [149]. However, Karimi et al., showed that ICAM levels were higher in patients with type 2 DM compared to healthy patients and that higher levels were also associated with more microalbuminuria [150]. As one of the mechanisms of development of diabetic nephropathy is endothelial to mesenchymal transition, studies show that E-cadherin levels fall with more transition and loss of endothelium and could be considered a novel biomarker of diabetic nephropathy development and progression [151]. Similar findings with lower expression of E-cadherin were also observed in children with IgAN [152]. Further studies on diabetic nephropathy revealed that podocyte detachment in the early stage of the disease is mediated through α3β1-integrin [153]. Also, the use of integrin antagonists in a rat model of diabetic nephropathy suggested that treatment with MK-0429 might have a meaningful impact on proteinuria and prevent fibrosis in diabetic nephropathy [154].

## 5. Conclusions

Studies done in recent years have deepened our knowledge of CKD pathogenesis from development, through progression, to ESRD. This area has truly become a vast field of molecules connecting renal cells through intracellular signaling and mediating inflammation as the hallmark of CKD. Nevertheless, new molecules are being identified and knowledge on their interplay is still in its infancy. Additionally, oxidative stress plays an enormous role in atherogenesis and is a topic for itself. We have already elucidated some possible treatment options that mainly focus on inhibition of inflammation markers and have been done in the past years, however, a broad field of possibility remains and further insight into inflammation and atherogenesis in CKD will be gained once some of them come into clinical practice.

Individual references mentioning the selected group of inflammation markers can be viewed in Table 1.

## Figures and Tables

**Figure 1 biomedicines-09-00182-f001:**
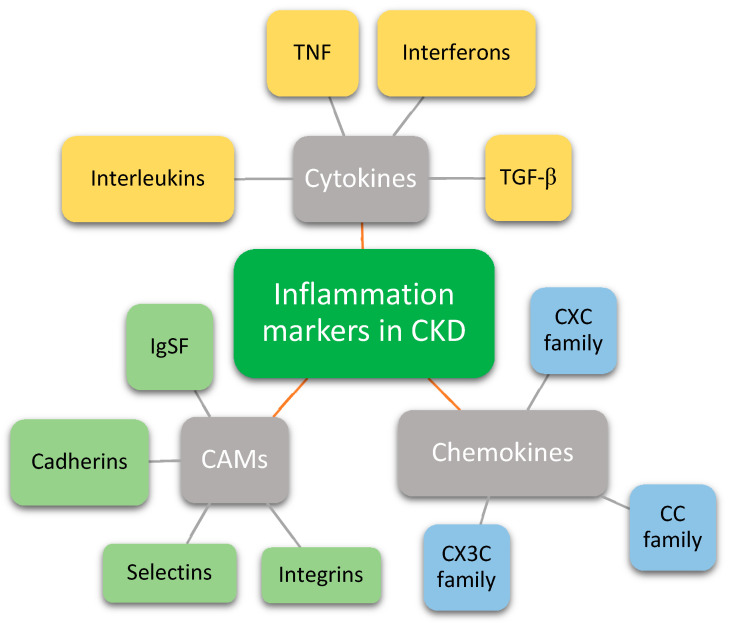
Groups of inflammation markers that have a role in chronic kidney disease. CKD—Chronic kidney disease; TNF—Tumor necrosis factor; TGF-β—Transforming growth factor beta; CAMs—Cell adhesion molecules; IgSF—Immunoglobulin superfamily.

**Table 1 biomedicines-09-00182-t001:** Reference list of articles after 2015 mentioning individual inflammation markers.

Inflammation Marker	Original Article	Review/Meta-Analysis/Case Report
IL 6	[15,16,17,19,20,22,23,25,27,30,34,145]	[8,18,36]
IL 1	[15,16,20,23,30,31,32,33,34,45,47,51]	[9,28,29,35,36,39,41,42,43,46,49,50,64]
IL 18	[23,51,122]	[9,28,35,36,37,38,39,41,42,46,49]
IL 17	[51,55,60]	[13,54,56,57,58,59,64]
IL 10	[20]	[11,61,62,63,64,65]
**Inflammasome**	[33,34,45,47,51,53]	[39,40,41,42,43,44,46,49,50,52]
TNF	[15,16,20,23,34,51,69,70,71,72,73,74,75,76,77,78,79,80,145]	[36,63,64,66,67,68]
IFN	[83,85,86,90]	[63,64,81,84,87,88]
TGF	[15,20,47,53,55,99,100,101,102,103,104,115,138,145,154]	[41,63,64,91,92,93,94,95,96,97,98,129]
CXC	[108,111,112,115,116,117]	[106]
CC	[51,120,121,122,126,127,130,131,139,145,147]	[36,59,63,106,123,124,129,132]
CX3C	[134,135,136,137,138,139]	[106,133]
CAMs	[51,142,145,146,147,148,149,150,151,152,153,154]	[93,140,141,143]

IL—interleukin; TNF—tumor necrosis factor; IFN—interferon; TGF—transforming growth factor; CAMs—cell adhesion molecules.

## Data Availability

Not applicable.

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
