# Peer review of "(untitled)"

_biomedicines, 2021, doi:10.3390/biomedicines9020182_

Round 1

Reviewer 1 Report

From a general point of view, my main concern is that the review is "too ambitious": it has a much too narrow field of interest. Throughout the review, loads of pathways (and the inflammation in CKD) are described. While the intent of that is concise and lacks a lot of the latest information: (i) the review is too short, making it uninspiring for the reader; (ii) the authors remain too superficial with regard to all the notions they try to summarize: every paragraph sounds, in the end, similar to the others: a short introduction on the pathway, a few arguments of how it may be implicated in CKD, and a closing remark. This manuscript cannot get more novel and comprehensive information, and the scope of obtaining new knowledge is still limited.

Author Response

Response to Reviewer #1:

From a general point of view, my main concern is that the review is "too ambitious": it has a much too narrow field of interest. Throughout the review, loads of pathways (and the inflammation in CKD) are described. While the intent of that is concise and lacks a lot of the latest information:

  1. The review is too short, making it uninspiring for the reader;

Answer: We thank Reviewer #1 for this comment. We have expanded the manuscript hoping it will now prove to be more inspiring for the reader. Additionally, we have added a table with recent references regarding individual inflammation markers.

  1. The authors remain too superficial with regard to all the notions they try to summarize: every paragraph sounds, in the end, similar to the others: a short introduction on the pathway, a few arguments of how it may be implicated in CKD, and a closing remark. This manuscript cannot get more novel and comprehensive information, and the scope of obtaining new knowledge is still limited.

Answer: We thank Reviewer #1 for this comment. We have deepened the information provided in the manuscript with its expansion. We would additionally like to explain that the manuscript design was chosen to enable the reader an introduction into each topic, how the topic is clinically relevant in chronic kidney disease, and whether any treatment options exist. We have focused on recent and novel literature, where most of citations range from 2015 to 2020, which is also evident in the new table provided. We acknowledge that this is not a molecular pathway review, where a deep description of each pathway could be its own review but, we hope, it would serve as a general review on introduction into clinical use or further research of inflammatory markers in chronic kidney disease.

Response to Reviewer #1:

From a general point of view, my main concern is that the review is "too ambitious": it has a much too narrow field of interest. Throughout the review, loads of pathways (and the inflammation in CKD) are described. While the intent of that is concise and lacks a lot of the latest information:

  1. The review is too short, making it uninspiring for the reader;

Answer: We thank Reviewer #1 for this comment. We have expanded the manuscript hoping it will now prove to be more inspiring for the reader. Additionally, we have added a table with recent references regarding individual inflammation markers.

  1. The authors remain too superficial with regard to all the notions they try to summarize: every paragraph sounds, in the end, similar to the others: a short introduction on the pathway, a few arguments of how it may be implicated in CKD, and a closing remark. This manuscript cannot get more novel and comprehensive information, and the scope of obtaining new knowledge is still limited.

Answer: We thank Reviewer #1 for this comment. We have deepened the information provided in the manuscript with its expansion. We would additionally like to explain that the manuscript design was chosen to enable the reader an introduction into each topic, how the topic is clinically relevant in chronic kidney disease, and whether any treatment options exist. We have focused on recent and novel literature, where most of citations range from 2015 to 2020, which is also evident in the new table provided. We acknowledge that this is not a molecular pathway review, where a deep description of each pathway could be its own review but, we hope, it would serve as a general review on introduction into clinical use or further research of inflammatory markers in chronic kidney disease.

Reviewer 2 Report

Petreski et al. offer a review on the role of inflammation in Chronic Kidney Disease (CKD).
The authors chose to focus the review on cytokines, tumor necrosis factor, interferon, and transforming growth factor, chemokines, and cell adhesion molecules. Some changes are needed to strengthen the review.

- CKD should be better defined according to the KDIGO classification in the introduction.
- the cardio-renal syndrome, cited in line 113, could be better defined.
- It would have been interesting to have a more detailed figure to present the groups of inflammation mediators. This would allowed a better understanding of  interaction of all these molecules and their effects.
- Even if the authors focused on mediators mentioned above, it would be interesting to add a section on oxidative stress and uremic toxins to understand the role of these common other effectors on inflammation.
- The inflammasome, which is increasingly studied in kidney disease, deserves to be more detailed in this review.

Author Response

Response to Reviewer #2:

Petreski et al. offer a review on the role of inflammation in Chronic Kidney Disease (CKD). The authors chose to focus the review on cytokines, tumor necrosis factor, interferon, and transforming growth factor, chemokines, and cell adhesion molecules. Some changes are needed to strengthen the review.

  1. CKD should be better defined according to the KDIGO classification in the introduction.

Answer: We thank Reviewer #2 for the overall positive feedback and constructive comments. We have defined CKD according to the KDIGO classification.

  1. The cardio-renal syndrome, cited in line 113, could be better defined.

Answer: We thank Reviewer #2 for this comment. We have reviewed the article from Buckley et al again. They have not defined their studied population as patients with cardio-renal syndrome but with heart failure and chronic kidney disease, thus we have changed this accordingly. As this is the only instance of mentioned cardio-renal syndrome in the manuscript, we have decided against defining the condition as it will cause unnecessary broadening of the manuscript.

  1. It would have been interesting to have a more detailed figure to present the groups of inflammation mediators. This would allow for a better understanding of interaction of all these molecules and their effects.

Answer: We thank Reviewer #2 for this comment and absolutely agree with him. Although for developing such a figure, the manuscript would have to focus on the molecular and not so much on the clinical effect of inflammation mediators. Additionally, this would impose to describe several other molecules and define inflammation pathways in a deeper manner, which would again lead the manuscript astray from the clinical use of the described inflammation markers in chronic kidney disease. Therefore, we have decided to only design a figure of the groups of inflammation markers mentioned in the manuscript, while keeping in mind that several others not mentioned here exist as well. Additionally, we have developed a table with individual references describing the mentioned inflammation markers for easier navigation and possible further reading of the reader.

  1. Even if the authors focused on mediators mentioned above, it would be interesting to add a section on oxidative stress and uremic toxins to understand the role of these common other effectors on inflammation.

Answer: We thank Reviewer #2 for this comment. We have discussed about including oxidative stress markers into the review. However, they represent a very broad topic, suitable for a review of their own, which is why we have decided against it. A review about them in diabetic nephropathy was done recently by our colleagues Vodošek Hojs et al in Antioxidants 2020, 9, 925; doi:10.3390/antiox9100925.

  1. The inflammasome, which is increasingly studied in kidney disease, deserves to be more detailed in this review.

Answer: We thank Reviewer #2 for this comment. We have created a section about the inflammasome (2.1.3.), as we have already mentioned it previously, and further expanded it.

Round 2

Reviewer 1 Report

The authors addresssed all questions, no further comments